# Effect of Oxyfluorination of PFA-Coated Metal Mesh with Superhydrophobic Properties on the Filtration Performance of SiO_2_ Microparticles

**DOI:** 10.3390/molecules28073110

**Published:** 2023-03-30

**Authors:** Kyung-Soo Kim, Cheol-Hwan Kwak, Seong-Min Ha, Jae-Chun Ryu, Young-Seak Lee

**Affiliations:** 1Department of Chemical Engineering and Applied Chemistry, Chungnam National University, Daejeon 34134, Republic of Korea; 2Carbon Composite Materials Research Center, Gumi Electronics & Information Technology Research Institute, Gumi 39171, Republic of Korea; 3Institute of Carbon Fusion Technology (InCFT), Chungnam National University, Daejeon 34134, Republic of Korea

**Keywords:** superhydrophobic, surface modification, metal mesh, PFA, oxyfluorination, water permeability

## Abstract

Recently, semiconductor wastewater treatment has received much attention due to the emergence of environmental issues. Acid-resistant coatings are essential for metal prefilters used in semiconductor wastewater treatment. Perfluoroalkoxy alkane is mainly used as an acid-resistant coating agent, since PFA has inherent superhydrophobicity, water permeability is lowered. To solve this problem, the surface of the PFA-coated metal mesh was treated via an oxyfluorination method in which an injected mixed gas of fluorine and oxygen reacted with the surface functional groups. Surface analysis, water contact angle measurement, and water permeability tests were performed on the surface-treated PFA-coated mesh. Consequently, the superhydrophobic surface was effectively converted to a hydrophobic surface as the PFA coating layer was surface-modified with C-O-OH functional groups via the oxyfluorination reaction. As a result of using simulation solutions that float silica particles of various sizes, the permeability and particle removal rate of the surface-modified PFA-coated stainless-steel mesh were improved compared to those before surface modification. Therefore, the oxyfluorination treatment used in this study was suitable for improving the filtration performance of SiO_2_ microparticles in the PFA-coated stainless-steel mesh.

## 1. Introduction

Technological alternatives to solve environmental problems caused by various industrial wastewaters have received much attention and been the main subject of many recent studies on environmental issues [1,2,3,4]. In particular, the problem of treating semiconductor wastewater, which is rapidly increasing due to advanced industrial development and the explosion of electronic products, has been recognized as an urgent problem [5]. The treatment of semiconductor wastewater must be carefully handled because of the specific characteristics of each process. For example, since semiconductor wastewater contains a high concentration of strong acids such as hydrofluoric acid for etching, an acid-resistant system is required for the treatment of this wastewater [6,7,8].

In general, a mesh-type prefilter is used for pretreatment to remove suspended particles in wastewater, and a metal material is preferred for mechanical durability [9]. When a metal prefilter is used in a strongly acidic environment such as the semiconductor wastewater mentioned above, it is essential to modify the metal surface to impart acid resistance. Various methods, including a coating method, are known for surface modification of a metal mesh to provide acid resistance, and fluorine-based polymers such as polytetrafluoroethylene (PTFE), perfluoroalkoxy alkane (PFA), and polyvinylidene fluoride (PVDF), which have high acid resistance, are widely applied as coating materials [10,11,12]. The Jiang research team manufactured the membrane using PVDF and used it to sepa-rate oil and water with superhydrophobic properties [13]. As such, the intrinsic fluorine-based polymer is expected to have low water permeability due to superhydrophobic properties, and thus wastewater treatment efficiency is expected to be reduced. Therefore, it is necessary to improve water permeability by modifying the metal mesh coated with the fluorine-based polymer surface.

Among several methods, the chemical wet method [14,15,16,17,18] and the plasma dry method [19,20,21,22,23,24,25] have been mainly studied to increase the water permeability of surfaces coated with fluorine-based polymers. The Vankelecom research team showed improved permeability by modifying the surface of the PVDF-coated membrane through crosslinking with para-xylenediamine (XDA) [26]. The Okubo research team improved the adhesion characteristics of PTFE-coated surfaces through a thermal plasma method using He gas [24]. The chemical wet method has some disadvantages in that the treatment process is complicated and time-consuming [27]. However, in the case of the plasma dry method, the process is relatively simple, and the treatment time is short. After a certain period of treatment, it is difficult to recover the original surface characteristics and completely convert the surface. [28].

To overcome these disadvantages, an oxyfluorination method was proposed as an alternative for effective surface modification [29]. In the oxyfluorination method using a mixed gas of fluorine and oxygen, fluorine radicals due to their high reactivity can generate peroxy radicals, and hydrophilic functional groups such as C-O-OH functional groups are formed through the radical reaction with original surface functional groups [30,31,32]. The Lee research team improved adsorption performance by modifying the surface through oxygen-containing fluorination of activated carbon fibers [33]. A surface modified by oxyfluorination can form a permanent hydrophilic covalent bond on the surface of a fluorine-based polymer, unlike a plasma method; therefore, the water permeability is improved and maintained for a long period [34].

In this study, stainless-steel meshes with various open pore sizes were coated with perfluoroalkoxy (PFA) polymer, known as a fluorine-based polymer, to provide acid resistance. The coating surface was then modified from a superhydrophobic surface to a hydrophobic surface using an oxyfluorination reaction. The surface modification before and after the oxyfluorination reaction was observed by measuring the water contact angle and surface energy. The permeability improvement was confirmed by the permeability test, and optimal conditions for surface modification via oxyfluorination were derived. Finally, the improvement of filtration efficiency as a prefilter was examined by removing suspended particles in water.

## 2. Results and Discussion

### 2.1. Morphological Changes on the Surface of the PFA-Coated Mesh

The SEM image in Figure 1 shows the changes in the surface morphology of the stainless-steel mesh before and after PFA coating and surface treatment. In the case of stainless-steel mesh, the open pore size was approximately 26, 34, and 38 μm, and the surface was smooth (see Figure 1a–c). After the application of the PFA coating, the pore size decreased to approximately 23, 32, and 36 μm due to PFA coating thicknesses of 1 to 1.5 μm. In addition, it was confirmed that the surface of the stainless-steel mesh after PFA coating was rougher than the original surface. However, it was confirmed that a uniform PFA coated layer on the surface of the stainless-steel mesh was achieved through the pore not being blocked by the spraying method (see Figure 1d–f). Particularly, there was no significant difference in the PFA coated surface of the stainless-steel mesh before and after the oxyfluorination treatment as shown in Figure 1g–i. Surface treatment with other chemicals or plasma may be accompanied by surface damage, which may affect durability. Therefore, these results showed that surface treatment through oxyfluorination is an appropriate way to modify surface properties while minimizing surface damage [35,36,37]. In addition, a corrosion test was conducted using hydrofluoric acid for 30 days to confirm whether the mesh had acid resistance by PFA coating. As a result of visually checking the degree of corrosion at the end of the experiment, corrosion of the steel mesh was not observed in both the PFA-coated mesh or the oxyfluorinated sample. Figure 2 shows distribution of oxygen and fluorine elements in stainless- steel mesh before and after PFA coating and oxyfluorinated surface treatment. It was confirmed that the PFA coated layer was well formed by the spraying method as oxygen and fluorine of the PFA coated stainless-steel mesh were uniformly distributed on all surfaces (see Figure 2a). In addition, it was confirmed that the surface element distribution of the stainless-steel mesh coated with PFA after oxyfluorination treatment was similar to the stainless-steel mesh before oxyfluorination treatment. Through this, it was confirmed that surface treatment through oxyfluorination is a method that can modify surface properties while minimizing surface damage (see Figure 2b,c). In addition, despite the increase in the oxygen gas volume ratio during oxyfluori-nation, it is similar to the untreated PFA-coated stainless-steel mesh, but the blue color of the SEM-EDS photograph of O50-PFA/mesh is observed, which clearly increases the distribution of oxygen elements (see Figure 2d).

### 2.2. Surface Analysis

To confirm the changes in the chemical bonds via oxyfluorination treatment, XPS was performed on PFA-coated stainless-steel mesh before and after the treatment. Figure 3 shows the results of XPS analysis with oxygen content introduced by oxyfluorination. Since no Fe signal was detected on the coated surface, the coating layer fully covered the stainless-steel mesh surface with PFA. C1s, O1s, and F1s peaks were identified at 284.5, 485.0, and 686.0 eV, respectively (see Appendix A). The O1s peak intensity increased with an increase in the oxygen gas volume ratio during oxyfluorination. Deconvolution peaks of C1s spectra were obtained for more detailed information on the surface chemistry. The C-O bond at a binding energy of 286.0 eV was due to the alkoxy bond that existed in the PFA molecular structure. The C-O bond appearing at 286 eV increased with increasing oxygen gas fraction. This result suggested that new C-O bonds were created by the oxyfluorination reaction. Each element ratio calculated from the XPS peak area is displayed in Table 1. The oxygen atom ratio increased as the fraction of oxygen gas increased compared to that before oxyfluorination treatment. This result was potentially caused by the formation of oxygen-containing functional groups on the surface of the mesh coated with PFA through thermal oxyfluorination treatment. 

FT-IR was also used to confirm the change in the chemical structure of the PFA coat-ing layer. Appendix A shows an FT-IR spectrum of a PFA coated mesh with an oxygen content introduced by oxyfluorination. It is confirmed that the C-O bond of 1000 to 1300 cm^−1^ increases as the partial pressure of gas in oxyfluorination increases. This suggests that the increase in the oxygen gas fraction introduced by oxyfluorination reacts with the active point generated by F_2_ gas to form peroxy radical. The characteristic peak due to C=O binding at 1703 cm^−1^ appeared only in the O50-PFA/mesh treated sample. These results show that a lot of peroxy radicals are formed by many oxygen gas fractions, increasing the functional group of C-O-OH, and oxidizing in the air to form C=O bonds. The surface modification of the PFA-coated stainless-steel mesh by oxyfluorination may be described as shown in Figure 4. Fluorine and oxygen gas are introduced into the reactor during oxyfluorination treatment, and fluorine radicals are formed when the temperature rises. An active point is formed on the surface of the PFA coated on the stainless-steel mesh by the fluorine radical thus formed. They attack the C-O-C bond of the PFA molecules, and react with oxygen gas inside the reactor to produce peroxy radicals. The resulting peroxy radical reacts in the air to form a C-O-OH bond. The formed C-O-OH functional group may provide surface hydrophobicity of the PFA coated mesh having surface superhydrophobic properties due to C-F binding. 

### 2.3. The Change in the Water Contact Angle

The water contact angle was measured to determine the surface change due to the introduction of peroxy functional groups on the surface by oxyfluorination. The water contact angle with the variation of the oxygen gas fraction is displayed in Figure 5a. The water contact angle of the original stainless-steel mesh greatly increased from 116° to 150° surface, as predicted in the chemical bond structure of the PFA material. However, it was observed that the contact angle decreased to 120° after the oxyfluorination reaction, and the extent of the decrease in the water contact angle increased as the fraction of oxygen gas used in the reaction increased. These results indicated that the change in the water contact angle before and after the oxyfluorination reaction was influenced by the change in the chemical bonding structure of the surface rather than due to the change in morphology, such as the change in roughness due to surface etching. Meanwhile, the contact angle was additionally measured with diiodomethane to determine the change in the surface energy of the PFA-coated mesh by oxyfluorination, as shown in Figure 5a. As shown in Figure 5b, the surface energy increased significantly after the oxyfluorination treatment. It was confirmed that the surface energy of PFA-coated stainless-steel mesh increased 1.21 times from 16.58 to 20.18 mN.m^−1^ through oxyfluorination, and the surface energy tended to increase as the oxygen partial pressure of oxyfluoride increased. These results indicated that the decrease in water contact angle after oxyfluorination treatment was due to surface modification of superhydrophobic to hydrophobic as surface energy increased. The Belov research team improved the operating characteristics of polymer materials through surface modification through fluorination. Through the oxyfluorination method, surface modification was performed from 117.3° to 96.8°, and surface energy was in-creased from 13 to 25 mJ/m [38].

### 2.4. Water Permeability and Particle Removal Rate

To check whether the water permeability of the PFA-coated mesh was improved by oxyfluorination, a permeability test using a gravity-driven filtering device was performed, and the results are shown in Figure 6a. In the case of the stainless-steel mesh filter, the time it took for the water to drain all the water due to gravity was 8 s (black line). However, in the case of the PFA-coated stainless-steel mesh filter, it took more than 20 min to drain all the water (red line). Moreover, in the case of the PFA-coated stainless-steel mesh filter treated with oxyfluorination, 9 min and 40 s were consumed (blue line). This result indicates that the PFA-coated stainless-steel mesh was successfully modified from superhydrophobic to hydrophobic. A particle removal experiment was performed with a simulated solution in which silica particles were suspended to examine the applicability in wastewater.

Figure 6b compares filtration efficiency through samples prepared with particles in the range of 5 to 55 μm of suspended SiO_2_ microparticles. A dispersion solution of particles of various sizes is filtered on a mesh prefilter, and a solvent before and after the filtration is compared to show the filtration efficiency. It can be found that the filtration efficiency of the stainless-steel mesh increases as the size of the SiO_2_ particles increases. It is observed that the filtration efficiency of SiO_2_ particles of 35 μm or less is low because the pore size of stainless-steel mesh is 34 μm. In addition, it can be seen that the PFA-coated stainless-steel mesh has improved filtration efficiency than the stainless-steel mesh. This result is potentially due to the reduced pore size of the stainless-steel mesh due to the coating thickness of PFA. It may be seen that as the pores of the stainless-steel mesh are controlled through the PFA coating, the filtration efficiency of SiO_2_ particles smaller than the pore size is increased by about two times. Therefore, the PFA coating plays a positive role in improving filtration efficiency as well as improving acid resistance. It was confirmed that the filtration efficiency increased as oxyfluorination was performed on the PFA-coated stainless-steel mesh. The filtration efficiency is improved as the oxygen gas fraction increases in the oxygen fluoride reaction. These results are believed to be due to the increased oxygen content of the PFA-coated stainless-steel mesh, which forms a C-O-OH functional group, converting it from a superhydrophobic surface to a hydrophobic surface, improving the water permeability.

Figure 6c shows the particle size distribution before and after filtering the produced SiO_2_ microparticles dispersion solution with PFA coating and oxyfluorinated stainless- steel mesh. Three large peaks are observed in the dispersion solution of the unfiltered SiO_2_ microparticles (black). The three peaks show microparticles of less than 1 μm, 1 to 35 μm microparticles, and more than 35 μm microparticles. When the PFA-coated stainless-steel mesh is filtered through a filter, two peaks are observed and it is determined that 35 microparticles were completely filtered. It can be seen that 35 μm SiO_2_ microparticles were completely removed because the pores of the PFA-coated stainless-steel mesh were 32 μm. The O50-PFA/mesh sample also has two peaks after filtering the dispersion solution, which can be found to be similar to the graph with PFA/mesh. Through this, pores are greatly involved in the removal of SiO_2_ microparticles, but it is judged that filtration efficiency is improved due to an increase in permeability through oxyfluorination.

As shown in Figure 6d, the permeability of the stainless-steel mesh decreased after PFA coating but increased again after oxyfluorination treatment. The coating of PFA on stainless-steel mesh shows a low permeability of 2.92 × 10^−13^ m^2^, up to 1.12 times the improvement of 3.26 × 10^−13^ m^2^ after oxyfluorination treatment. The Jia research team studied the water permeability of various wood biocarbons and confirmed that the water permeability in the radial direction was up to 2.5× 10^−13^ m^2^ [39]. Through this, it is judged that the PFA-coated stainless-steel mesh used in this experiment has the advantage of improving permeability through simple surface treatment. The Liu research team modified the surface with nano-porous TiO_2_ to improve permeability and showed an improved permeability of 1.21 times [40]. Oxyfluorination is an effective surface treatment method because it shows a similar improvement effect with a simple surface treatment method, although it shows a lower permeability improvement rate than the previous experimental results. As seen from the pressure drop results shown on the right side of Figure 6d, the pressure drop decreased as the oxygen gas fraction increased, which was most likely due to the increase in permeability. As described above, as a higher fraction of oxygen gas was used in the oxyfluorination reaction, more C-O-OH functional groups were formed on the surface to modify the surface from superhydrophobic to hydrophobic. This result strongly suggested that the reduced water permeability by PFA coating for corrosion resistance to strong acids could be recovered by an oxyfluorination treatment.

## 3. Materials and Methods

### 3.1. Materials

The PFA coating solution used for the coating was purchased from a commercially available PFA coating solution (37.5 wt%, 858G-210, Chemours Co., Wilmington, DE, USA) and was used without any further treatment. The stainless-steel mesh (Arim Co., Seoul, Korea) was a twill mesh with aperture sizes (number of apertures per inch) of 26 (#500), 32 (#400), and 38 (#350) μm. Each mesh was ultrasonically cleaned in ethanol and acetone for 10 min each to remove contaminants remaining on the surface before coating. For the oxyfluorination reaction, high-purity fluorine gas (99.0%, Messer Grieheim GmbH, Bad Soden, Germany) and high-purity oxygen gas (99.999%) were used. Silicon dioxide microparticles (SiO_2_, purity of 99.5%, Sigma Co., Seoul, Korea) used a particle size of ~325 mesh, and 40–75 μm of SiO_2_.

### 3.2. Stainless-Steel Mesh Coating with PFA Polymer

The PFA polymer coating solution was dispersed by stirring at 150 rpm for 15 min at room temperature. Then, 0.3 mL of the PFA dispersed coating solution was sprayed on a stainless-steel mesh (80 mm× 80 mm) at a distance of 10 mm and under a spraying pressure of 8 bar. The spray-coated stainless-steel mesh was dried for 10 min at 150 °C in an air atmosphere and then heat treated at 400 °C for 15 min in an air atmosphere to fuse the coated PFA polymer onto the mesh surface. The coated sample was named PFA/mesh.

### 3.3. Oxyfluorination Reaction

The oxyfluorination reaction of the PFA-coated stainless-steel mesh was carried out using a batch-type device equipped with a nickel reactor. The detailed oxyfluorination procedure was presented in our previous work [41,42,43,44,45]. Before oxyfluorination, the reaction chamber was evacuated three times using a pressure-reducing pump and purged by nitrogen gas to remove the residual moisture. The reactor in a vacuum state was heated to 300 °C at a rate of 10 °C/min. The mixed gases with fluorine and oxygen gas were injected into the reaction chamber at partial pressure ratios of F_2_:O_2_ = 9:1, 7:3, and 5:5 and were then maintained for 20 min. After the reaction was completed, the inside of the reactor was evacuated three times using nitrogen gas to remove the unreacted fluorine gas, and then the sample was collected. The obtained samples were denoted O10-PFA/mesh, O30-PFA/mesh, and O50-PFA/mesh according to the mixed oxygen gas fraction.

### 3.4. Surface Energy

The surface energy was calculated by using the Owens–Wendt model derived from several equations and contact angles measured with water and diiodomethane. The surface energy was calculated through Young’s equation [46,47] with the contact angle (*θ*) formed by the liquid droplet on the PFA-coated mesh as shown below:(1)$$\gamma_{s}-\gamma_{sl}=\gamma_{l}\cos\theta \;\mathrm{Young}’s\;\mathrm{equation}$$
where $\gamma_{s}$ is the surface free energy of a solid, $\gamma_{sl}$ is the solid–liquid interfacial tension, and $\gamma_{l}$ is the surface tension of the liquid. An interfacial adhesion between solid and liquid is used for the characterization of intermolecular forces across the interface. The work of adhesion ($W_{sl}$) represents [46,47] these forces and is calculated using interfacial energy components as follows.
(2)$$W_{sl}=\gamma_{l}+\gamma_{s}-\gamma_{sl}\;\mathrm{Work}\;\mathrm{of}\;\mathrm{adhesion}$$

Combining Equation (2) with Young’s equation (Equation (1)) gives the Young-Dupré equation [48] in terms of *θ* and $\gamma_{lv}$ as follows.
(3)$$W_{sl}=\gamma_{l}\left(1+\cos\theta \right)\;\mathrm{Young}\text{-}\mathrm{Dupre}\;\mathrm{equation}$$

Good’s equation [49] is introduced to separate nonpolar dispersive and polar components of surface energies as follows:(4)$$\gamma_{sl}=\gamma_{s}-\gamma_{l}-2{\left(\gamma_{s}^{d}\cdot \gamma_{l}^{d}\right)}^{\frac{1}{2}}-2{\left(\gamma_{s}^{P}\cdot \gamma_{l}^{P}\right)}^{\frac{1}{2}}\;\mathrm{Good}’s\;\mathrm{equation}$$
where $\gamma_{s}^{d}$ and $\gamma_{l}^{d}$ represent the dispersive components of the surface energy for solid and liquid, respectively, and $\gamma_{s}^{P}$ and $\gamma_{l}^{P}$ are the polar components of the surface energy for solid and liquid, respectively. The Owens–Wendt equation [50,51] is finally derived by combining Good’s equation (Equation (4)) with the Young-Dupre equation (Equation (3)) as follows.
(5)$$\gamma_{l}\left(1+\cos\theta \right)=2{\left(\gamma_{s}^{d}\cdot \gamma_{l}^{d}\right)}^{\frac{1}{2}}+2{\left(\gamma_{s}^{P}\cdot \gamma_{l}^{P}\right)}^{\frac{1}{2}}\;\mathrm{Owens}–\mathrm{Wendt}\;\mathrm{equation}$$

By substituting the contact angles of nonpolar and polar solutions into Equation (5), the nonpolar energy and the polar energy of the sample can be calculated, and the sum of the nonpolar energy and the polar energy of the sample is expressed as the surface free energy [51].

### 3.5. A Test Method for Particle Removal Rate and Water Permeability

The experimental setup was prepared as shown in Appendix A to investigate the particle removal efficiency. As-prepared PFA/mesh and O50-PFA/mesh with a diameter of 80 mm were mounted on this setup. An aqueous SiO_2_ dispersion was prepared using deionized water (DI water) and SiO_2_ microparticles with a size distribution of ~75 μm. The aqueous SiO_2_ dispersion flowed through the mounted mesh. A total of 1 L SiO_2_ dispersion flowed through the mesh at a flow rate of 1 L/min for 1 min. The SiO_2_ dispersion samples were collected through two valves, which were installed in front and behind the mesh in the direction of fluid flow for analysis of the particle removal efficiency by the mesh. In order to calculate the filtration efficiency, the filtration efficiency was calculated by the following Equation (6). In addition, it was calculated using the following Equation (7) to obtain permeability in the mesh manufactured using the manufactured SiO_2_ dispersion sample.
(6)η=1−CfC0×100%
where $C_{0}$ and $C_{f}$ are the water concentrations of the solvent in which the original SiO_2_ microparticles are dispersed and the solvent after filtration, respectively. The water content before and after filtration was measured using a Karl Fischer Titrator.
(7)$$Q=\frac{k\times A\times \Delta P}{\mu \times L}\;\mathrm{Darcy}’s\;\mathrm{law}$$
where Q is Volumetric flow rate (m^3^ s^−1^), k is the permeability (m^2^) of the SiO_2_ dispersion collected from the filtration, and A is the Cross-sectional area (m^2^), ΔP is the Pressure difference (Pa), μ is the Dynamic viscosity (Pa s^−1^), and L is the Thickness (m).

### 3.6. Analysis

To observe the change in chemical properties and chemical bonding structure before and after PFA coating and oxyfluorination treatment, a sample was obtained by cutting some of the PFA-coated stainless-steel mesh surface and analyzed using X-ray photoelectron spectroscopy (XPS, K-Alpha XPS instrument, ThermoFisher Scientific, East Grinter, UK). A Vacuum Infrared spectrometer (FT-IR, VERTEX 80v, Bruker Co., Billerica, MA, USA) was used to identify changes in the surface chemical properties of PFA coating and oxyfluorination treatment. To determine the degree of hydrophobic change, the contact angle of each sample was measured using a contact angle device (Phoenix 300, Surface Electro Optics Co. Ltd., Suwon, Republic of Korea). DI water and diiodomethane solution were used to measure the contact angle. The measurement was performed at least 8 times, and an average value was obtained. The water and diiodomethane contact angle measurements were performed with 5 to 8 μL at least eight times, and an average value was obtained. To confirm the change in the morphologies and chemical composition of the coated surface before and after the oxyfluorination reaction, the samples were analyzed with field emission scanning electron microscopy (FE-SEM, Hitachi S-4800, Hitachi, Japan) and energy-dispersive X-ray spectroscopy (EDS).

## 4. Conclusions

In this study, oxyfluorination treatment was performed to improve the filtration performance of a stainless-steel mesh coated with PFA to impart acid resistance. It was found that C-O-OH functional groups were formed on the surface of the PFA-coated stainless-steel mesh treated by injecting the mixed fluorine and oxygen gas. And the contact angle results confirmed that the PFA-coated stainless- steel mesh with superhydrophobicity properties of 150° was converted into hydrophobicity properties of 120°. The C-O-OH functional group was formed on the surface of the PFA coated mesh treated by oxyfluorination treatment, indicating that the permeability increased by more than 1.12 times from up to 2.92 × 10^−13^ to 3.26 × 10^−13^ m^2^. Finally, the particle removal rate and permeability of PFA/mesh after oxyfluorination were improved by reducing the pressure drop of PFA/mesh after oxyfluorination treatment from 0.38 to 0.34 bar, compared to PFA/mesh before oxyfluorination treatment. Therefore, the oxyfluorination treatment used in this study was found to be suitable as a way to improve the permeability of PFA-coated stainless-steel mesh and the filtration of SiO_2_ microparticles

## Figures and Tables

**Figure 1 molecules-28-03110-f001:**
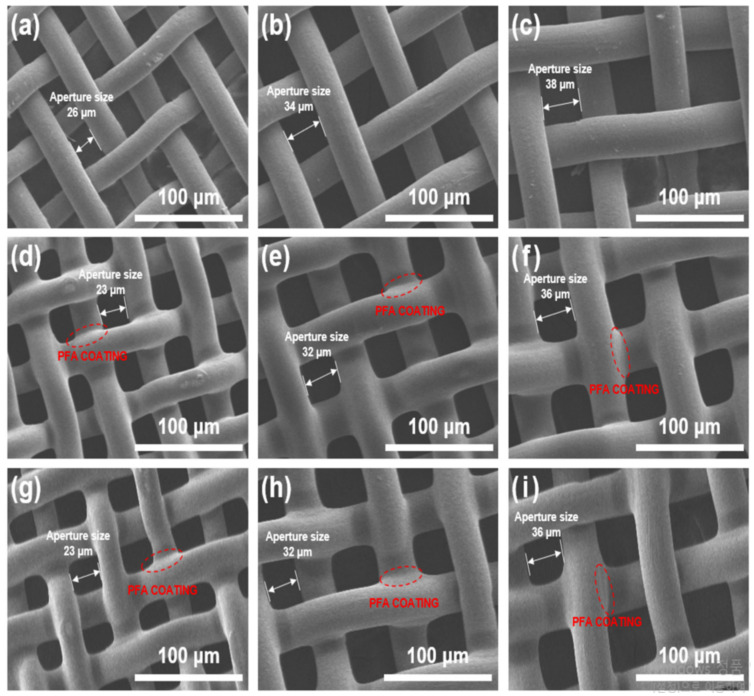
SEM images of the stainless-steel mesh (**a**) 26 μm, (**b**) 34 μm, (**c**) 38 μm and PFA coated stain-less-steel mesh (**d**) 23 μm, (**e**) 32 μm, (**f**) 36 μm and oxyfluorination treated PFA coated stainless-steel mesh (**g**) 23 μm, (**h**) 32 μm, (**i**) 36 μm.

**Figure 2 molecules-28-03110-f002:**
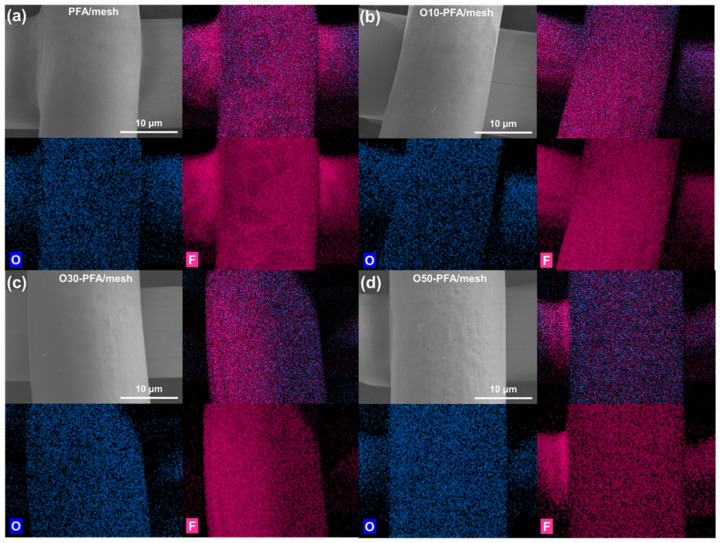
SEM–EDS images of (**a**) PFA-coated stainless-steel mesh, (**b**) O10-PFA/mesh, (**c**) O30-PFA/mesh and (**d**) O50-PFA/mesh.

**Figure 3 molecules-28-03110-f003:**
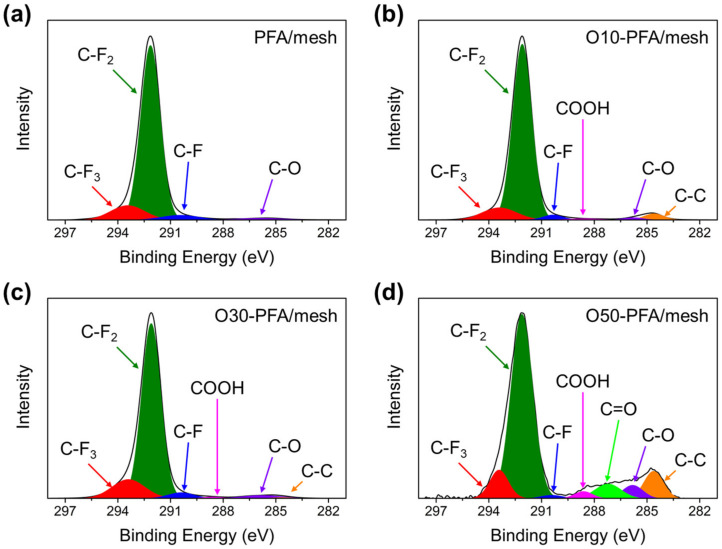
XPS spectra of stainless-steel mesh coated PFA before and after surface modification by oxyfluorination: (**a**) PFA/mesh, (**b**) O10-PFA/mesh, (**c**) O30-PFA/mesh, and (**d**) O50-PFA/mesh.

**Figure 4 molecules-28-03110-f004:**
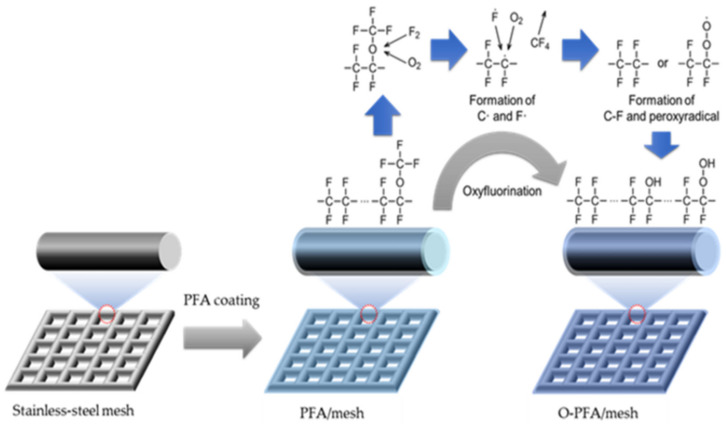
The proposed mechanism of surface modification on PFA-coated mesh by oxyfluorination.

**Figure 5 molecules-28-03110-f005:**
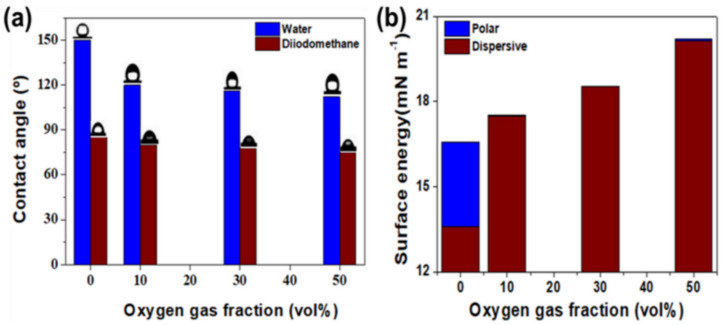
(**a**) Contact angle of water and diiodomethane with the oxygen gas fraction during oxyfluorination and (**b**) surface energy of the polar and dispersive components calculated from the contact angles.

**Figure 6 molecules-28-03110-f006:**
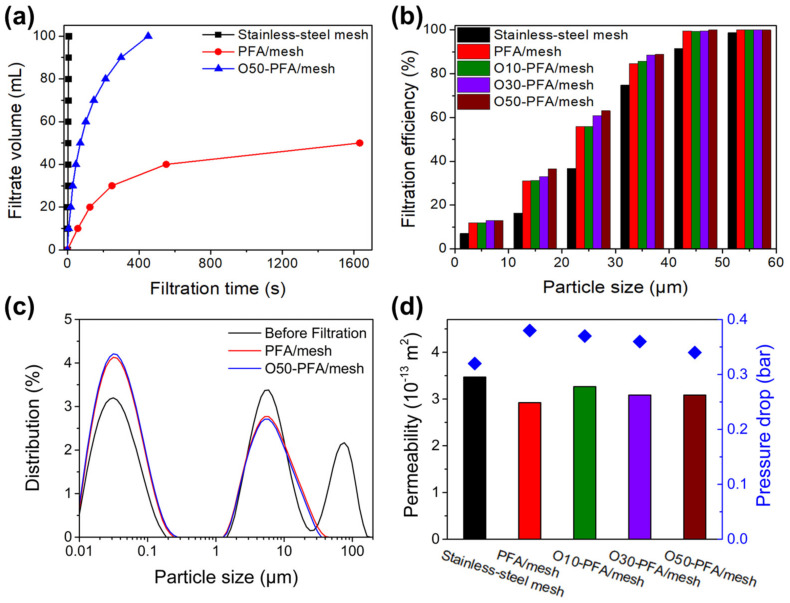
(**a**) The gravity-driven water permeation time of stainless-steel mesh, PFA/mesh, and O50-PFA/mesh. (**b**) The filtration efficiency in the segmented size range of the SiO_2_ particles under 60 μm, (**c**) Particle size distribution before and after filtration of SiO_2_ suspension using O50-PFA/mesh. (**d**) The pressure drop (blue diamond) during the filtration of SiO_2_ dispersion, and calculated permeability coefficients of stainless-steel mesh, PFA/mesh, and oxyfluorinated PFA/mesh.

**Table 1 molecules-28-03110-t001:** The atomic percentage of carbon and oxygen on the surface with the variation in the oxygen gas fraction in the oxyfluorination reaction.

Sample	Elemental Content (Atomic %)		
	C	O	F
PFA/mesh	31.5	0.4	68.1
O10-PFA/mesh	31.7	0.5	67.8
O30-PFA/mesh	31.4	0.5	68.1
O50-PFA/mesh	28.8	6.4	64.8

## Data Availability

Data are contained within the article.

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
