# Peer review of "Effect of Oxyfluorination of PFA-Coated Metal Mesh with Superhydrophobic Properties on the Filtration Performance of SiO2 Microparticles"

_molecules, 2023, doi:10.3390/molecules28073110_

Round 1

Reviewer 1 Report

This paper presents the preparation and characterization of acid-resistant coatings of stainless steel meshes (SSM), and testing of their ability to filter SiO2 particles of different sizes present in highly acidic wastewater produced by semiconductor industries. The acid resistant stainless steel meshes were prepared by spray coating clean SSM with PFA and subsequent treatment with oxyfluorinating reagent to reduce their hydrophobicity. This technology is important because it has applications in waste management of semiconductor industry. They have developed oxygenated-PFA coated stainless steel meshes which have superior performance for applications in removing SiO2 particles of different sizes from the wastewater. Here are some comments and suggestions:

1.       Figure 1 should include details about the treatments and mesh pore sizes.

2.       Figure 4 presents the mechanism of oxyflorination and formation of peroxide. Why the oxidation should stops with peroxide and not proceed to form carboxylic acid. XPS data in Figure 2 clearly showed the presence of O–C=O group. The FTIR data in Figure S3 also have peak at ca. 1500 cm-1 and hydrogen bonding F..O=C- is possible.  Explain.

3.       Figure 5 shows contact angle of 120 degree for the oxyfluorinated samples which is still highly hydrophobic. Why hydrophilic sample with contact angle below 90 degree was not prepared and tested. Explain.

4.       The most important measure of the mesh performance is the “Filtration efficiency” as presented in Figure 6, however it is not clearly defined in the manuscript. So, include a clear definition of the “Filtration efficiency”.

5.       Figure S4 has important information and should be included in Figure 6.

Author Response

Thank you for your effort.

Reviewer 2 Report

I have reviewed carefully the manuscript and found that there is room for improvement.

Abstract – Kindly highlight the key results in abstract. For instance, contact angle, filtration efficiency and permeability.

Introduction does not provide detailed review on the coated metal meshes and highlight the key findings from respective studies.

2.1 Materials – What is the concentration of PFA coating solution? Provide it! The information regarding SiO2 is not provided. Besides, please explain why 26, 32 and 38 meshes were selected for this work.

2.2 – Please justify why 0.3 mL and 10 mm distance were used in this work.

Results and discussion – I strongly advise the authors to compare their findings with other relevant studies instead of pure discussion based on their findings.

Figure 1 caption – Please provide full caption to explain each image (a-j). It is hard to get information from the current caption. Also, please provide dimension to justify the increase in coating thickness.

Figure 2 – Please mention the type of mesh used for this experiment.

Figure 3 vs Table 1 – The % for the element is quite confusing. I suggest the authors to be consistent when reporting the element %. Also, discuss if the increase in % correlate to the gas ratio.

Eq (1)-(5) should be arranged in Methodology section!

Figure 6 – The formula used to determine filtration efficiency and permeability are not provided. Besides, I strongly advise the authors to use a common unit for permeability. For instance, L/m2.h.

Conclusion – Key findings (in number) should be provided here.

Author Response

Thnak you for your review.
